# An Importance-Performance Analysis of Korean Middle School Students’ Health Management Awareness during the Post-COVID-19 Era Depending on Sex

**DOI:** 10.3390/healthcare12070763

**Published:** 2024-03-31

**Authors:** Chul-Min Kim, Yi-Hang Huang, Tong Zhou, Min-Jun Kim, Hyun-Su Youn

**Affiliations:** 1Department of Physical Education, Kyung Hee University, Yongin-si 17104, Republic of Korea; cmkim@khu.ac.kr; 2Department of Physical Education, Korea University, Seoul 02841, Republic of Korea; huang1994@korea.ac.kr (Y.-H.H.); zhou941002@korea.ac.kr (T.Z.); 3Department of Physical Education, Shin Han University, Uijeongbu 11644, Republic of Korea; 4Department of Physical Education, College of Education, Won Kwang University, Iksan-si 54538, Republic of Korea

**Keywords:** COVID-19 endemic phase, Korean middle school students, health awareness, importance-performance analysis (IPA)

## Abstract

This study aimed to conduct importance-performance analyses (IPAs) based on Korean middle school students’ health management awareness during the post-coronavirus disease 2019 (COVID-19) era. Data were collected from 867 Korean middle school students (13–15 years old) via online and offline surveys between May and June 2023. Frequency analysis, reliability analysis, IPA based on the entire student group, and IPA depending on sex were carried out with the collected data, which revealed the following. First, regardless of sex, the IPA results indicated that four factors of mental health were located in the third quadrant, with one factor of the same variable in the fourth quadrant. The three factors of disease management were located in the third quadrant. Regarding physical activity, two factors were located in the first quadrant, one in the second quadrant, and one in the third quadrant. Regarding sleep management, two factors were located in the second quadrant, one in the third quadrant, and one in the first quadrant. Regarding eating management, two factors were located in the third quadrant and one in the fourth quadrant. Regarding the social distancing variable, all four factors were located in the third quadrant. Regarding hygiene management, two factors were located in the first quadrant, one in the third quadrant, and one in the fourth quadrant. Furthermore, the IPA results indicated sex differences in regular sports and vigorous movement activities associated with physical activity. Additionally, a sex difference was observed in regular diet associated with eating management. This study proposed possible measures for encouraging middle school students to recognize the importance of health and increase their health-related performance during the COVID-19 endemic phase.

## 1. Introduction

Since the World Health Organization (WHO) declared the end to coronavirus 2019 (COVID-19) as a public health emergency in May 2022, the South Korean government has relaxed COVID-19 guidelines and implemented autonomous social distancing. Over the past two years, the COVID-19 pandemic has posed significant challenges across various fields, including the social, economic, cultural, educational, and industrial sectors. Beyond these difficulties, Korean people have adapted to changes and realized the value of daily life. However, one of the problems related to the COVID-19 pandemic was poor physical activity caused by a prolonged sedentary lifestyle, which may contribute to several mental and physical diseases. Multiple reports and media have warned against the consequences of insufficient physical activity in adolescents, specifically [1].

A previous study on physical activity among adolescents during the COVID-19 pandemic indicated that activity was restricted by the contactless school-education system implemented during the crisis [2]. As minors, adolescents were subject to greater social distancing measures than adults to protect them from the pandemic. Consequently, adolescents reported significant negative changes during the two years of the pandemic, such as depression, anxiety, social isolation, maladjustment, stress, skipping breakfast, and insufficient sleep [3,4,5]. Other studies have suggested that changes brought about by the COVID-19 pandemic, such as lifestyle adjustments, inappropriate nutritional conditions, addiction to media, and reduced physical activity, have negatively affected adolescents’ physical and mental health [6].

Another study on the physical activity of adolescents during the COVID-19 pandemic identified a difference between therapeutic and preventative health factors, reporting significantly lower levels of importance and performance of physical activities [7]. The COVID-19 pandemic has also resulted in a reduction in adolescent physical activity and an increased rate of obesity [7]. Regarding the physical activities of adolescents, sedentary activities have increased, whereas medium- and high-intensity physical activities have decreased [8,9]. Furthermore, studies showed that adolescents’ health-related quality of life (HRQoL), including sleep, eating habits, and hygiene, had decreased [10], which negatively impacted their lifestyles [11,12].

In summary, changes in physical activity, lifestyle habits, and social environment related to COVID-19 contributed to a decline in adolescents’ health levels [13]. The restrictions on physical activity caused by school closures and non-face-to-face classes during the pandemic may have been a reason for the worsening health status and awareness [13]. The new operation mode of school education during the COVID-19 pandemic negatively impacted adolescents’ physical health, learning of community-related capabilities, and academic achievement. Additionally, several measures of adolescents’ emotional health had low values [14].

Recently, the central and local governments in South Korea have implemented diverse physical exercise programs for citizens who lacked physical activity during the pandemic to restore their health [15]. The Ministry of Education and schools have worked to promote the physical activity of adolescents, in particular [16,17], based on reports [7,18] of insufficient physical activity among adolescents. However, there is little research that investigates and analyzes the actual, endemic situation by categorizing adolescents’ health awareness. According to previous research, adolescent health expands the scope of physical, mental, and social health (as presented by the WHO) to understand specific health-related awareness such as eating habits, sleep, and health, and determines adolescent health-related quality of life [10].

Under these circumstances, this study recognizes the need to examine the health conditions of adolescents, implement policies that promote physical education in schools, and encourage adolescent health recovery. Analysis of adolescent health management is expected to provide essential data for the design and implementation of long-term health recovery programs. This study investigates the perception of Korean middle school students’ healthcare behavior in the post-COVID-19 era, identifies trends through the analysis of the importance and practice of health care behavior, and ultimately derives implications for physical education. Based on the analysis, it identifies tendencies related to health management awareness.

This study used an IPA analysis, which is a model devised by Martilla and James [19]. It is based on a multi-attribute model and measures the importance and performance of an item and visualizes the results on a two-dimensional drawing based on the *X*-axis (performance) and *Y*-axis (importance).

The results of this analysis were compared with existing work on the health management awareness of Korean middle school students during the initial and prolonged COVID-19 pandemic phases to investigate the current conditions and tendencies of their health management awareness [3,11].

Furthermore, this study analyzed the middle school students’ perception of health and the characteristics of their physical activities [20,21,22] according to sex, given that multiple studies have reported sex as a core variable for understanding the characteristics of adolescents’ physical activities [23,24]. This study is a follow-up to previous work which found that physical activity, eating habits, and sleeping habits were among the poor health indicators during the early and sustained phases of the COVID-19 pandemic [11,25].

## 2. Methods

### 2.1. Participants

This study used middle school students living in South Korea as the study population. Participants were selected using convenience sampling, which is a non-probability sampling technique. Of the 882 respondents, 15 who indiscriminately marked their answers or had unclear answers were excluded. Consequently, this study selected 867 Korean middle school students in their first, second, or third year and aged 13–15 years. The selected participants completed online and offline surveys using Google Forms between May and June 2023. Table 1 presents the demographic characteristics of participants.

This study used middle school students living in Korea as its population. Participants were selected using convenience sampling, which is a non-probability sampling technique. The content validity of the measurement tool was verified by three professors majoring in sports and one doctor majoring in medical health. For further verification, a preliminary survey was conducted on 100 students in the first to third years of middle school in Korea. Based on the results of the preliminary survey, inappropriate items were removed.

This survey was conducted online using Google Forms for a total of one month, from May to June 2023, for first- to third-year middle school students in Korea. All questionnaire items were completed using self-assessment methods, and all questionnaire items except demographic characteristics were answered using a five-point Likert scale. In addition, to secure a larger number of samples, the questionnaire was distributed and collected through incumbent middle school teachers. In total, 882 responses were obtained; 15 incomplete or collectively completed were excluded, leaving 867 questionnaires as the final valid sample. Table 1 shows the demographic characteristics of the study participants.

Finally, we used G-power 3.1.6 [26] to calculate the number of samples, and the number of samples required for the independent *t*-test test was calculated. The standard was set to an effect size of 0.50, a significance level of 0.05, and a power of 0.95; the analysis revealed that 210 samples were required for this study.

### 2.2. Instruments

This study used scales proposed by existing studies that were consistent with its research purposes. Participants completed questions regarding their sex, experience with COVID-19 infection, and whether they wore a facial mask during physical activities; these items were measured as categorical variables. To measure the participants’ perception of health, this study used health perception scales developed by Ware [27]; these scales had their validity and reliability confirmed by Barakat et al. [28], Jones [29] and Lee et al. [30]. This study used seven variables: mental health, disease management, physical activity, sleep management, eating management, hygiene management, and social distancing. Questionnaire items were rated on a five-point Likert scale to calculate the scores for each item independently.

### 2.3. Reliability of Instruments

To confirm the reliability of the adopted scales, this study used an exploratory factor analysis to determine Cronbach’s α, a measure used to verify the internal consistency of items. The results of the analysis are listed in Table 2. As was observed in a previous study [7,11,25], seven variables with a factor loading of 0.5 or higher were derived, including eating management, disease management, mental health, physical activity, sleep management, hygiene management, and social distancing. The calculated Cronbach’s α ranged from 0.728 to 0.852, which ensured adequate reliability of the scales used in this study [31]. The calculation results are presented in Table 2.

### 2.4. Procedures and Data Analysis

The collected data were analyzed using SPSS (version 18.0; IBM Corp., Armonk, NY, USA). The detailed analysis procedure is as follows. First, EFA and reliability analysis were conducted based on the adjusted IPA. Second, an independent sample *t*-test was conducted to examine the difference in Korean middle school students’ healthcare perception based on sex. Third, to evaluate the importance and performance of each variable, three IPAs were conducted: one that considered sex, one that did not, and one for sex. For the revised IPA analysis, the satisfaction of the respondents by item was converted into natural log values and set as importance based on partial correlation analysis with overall performance. Finally, this study received ethical approval from the Wonkwang University IRB (WKIRB-202306-SB-041).

IPA analysis divides into four quadrants based on the center point of the the *X* and *Y* axes; data are given meaning based on the quadrant where they are located [19]. IPA analysis is used in various academic fields because it compensates for the problems of SWOT analysis that are subjectively evaluated by managers and managers [32], and it also helps field practitioners easily comprehend objective improvement measures [19].

In particular, as IPA analysis techniques gradually become commercialized, studies that critically consider traditional IPA analysis have been conducted. One representative problem is the basic statistical assumption that it should be linear and symmetrical between importance [33] and satisfaction [34], and this is not satisfied.

From this perspective, the best alternative to address traditional IPA challenges is the modified IPA analysis devised by Deng [35] to convert satisfaction with each attribute into natural logarithmic values. Importance can be derived through partial correlation analysis between overall satisfaction [36].

## 3. Results

### 3.1. Relative IPA of Health Management Awareness of Korean Middle School Students Regardless of Sex

The median (0.191) and mean (3.590) values of the average relative performance were established as standard nodes to form an IPA matrix based on the perceptions of Korean middle school students toward health management according to the factors. The calculation results are presented in Table 3 and Table 4 and Figure 1.

The details of the four quadrants of the IPA matrix for Korean middle school students, regardless of sex, are as follows.

The first quadrant was “keep up with good work”. In this quadrant, items showed high importance and performance, requiring their current conditions to be maintained. This quadrant comprised the following factors: happiness, interest in life, belongingness, and formation of a reliable relationship within the mental health variable; prevention of diseases, prescription management, and vaccination within the disease management variable; regular sports and vigorous movement activities within the physical activity variable; a stable sleep environment within the sleep management variable; and habitual hygienic activity and mandatory hygienic practice within the hygiene management variable.

The second quadrant was “concentrate here”. The items exhibited high importance and low performance, requiring urgent performance improvement. This quadrant comprised the following factors: daily gymnastic activity within the physical activity variable, regular sleep activity and appropriate sleep quality within the sleep management variable.

The third quadrant was “low priority”. The items showed low importance and performance, requiring greater effort than that exhibited currently. This quadrant includes the following factors: regular physical activity within the physical activity variable; appropriate sleep amount within the sleep management variable; regular diet and ideal eating habits for the eating management variable; the wearing of a facial mask in multi-use facilities, restricted access to multi-use facilities, practice of distancing, and use of a hand sanitizer for the social distancing variable; and preventing self-infection within the hygiene management variable.

The fourth quadrant was “possible overkills”. These items showed low importance and high performance, indicating that excessive effort was made on these items despite their low importance. This quadrant comprises the following factors: meaning of life within the mental health variable, appropriate diet amount for the eating management variable, and preventing the infection of others within the hygiene management variable.

### 3.2. Analysis of Middle School Students’ Performance Difference in Health Management Awareness Perception by Sex

An independent sample *t*-test was conducted to examine differences in performance regarding middle school students’ health management awareness by sex. The test results indicated statistically significant differences in physical activity, sleep management, and hygiene management. Male students performed better than female students in physical activity and sleep management variables while female students performed better in hygiene management. The calculation results are presented in Table 5.

### 3.3. Relative IPA of Korean Middle School Students’ Health Management Awareness by Sex

To form an IPA matrix based on the perceptions of Korean middle school students regarding health management awareness according to various factors, the following standards were applied. For male students, the median (0.201) and the median (3.6353) values of the average relative importance were established as standard nodes. For female students, the median (0.201) and the median (3.538) values of the average relative importance were established as standard nodes. The calculated results are presented in Table 6 and Table 7 and Figure 2.

The details of the four quadrants of the IPA matrix based on Korean middle school students by sex are as follows.

In the first quadrant (keep up the good work), where items show high importance and performance and require the maintenance of their current conditions, the following factors are displayed regardless of sex: happiness, interest in life, belongingness, the formation of a reliable relationship within the mental health variable; the prevention of diseases and prescription management within the disease management variable; stable sleep environment within the sleep management variable; activity of preventing the infection of others; habitual hygienic activity; and mandatory hygienic practice within the hygiene management variable. It should be noted that this quadrant contains factors specific to male students, such as regular sports and vigorous movement activities, within the physical activity variable.

In the second quadrant (possible overkill), where the items included show high importance and low performance and require urgent performance improvement related to them, the following factors are displayed regardless of sex: daily gymnastic activity within the physical activity variable and appropriate sleep quality within the sleep management variable. It should be noted that this quadrant contains factors specific to female students, such as regular sports and vigorous movement activities, within the physical activity variable.

In the third quadrant (low priority), where items showed low importance and performance and required greater effort than the present efforts, the following factors were displayed regardless of sex: regular physical activity within the physical activity variable; regular sleep activity, and appropriate sleep amount within the sleep management variable; ideal eating habits for the eating management variable; the wearing of a facial mask in multi-use facilities, restricted access to multi-use facilities, the practice of distancing, the use of a hand sanitizer for the social distancing variable; and preventing self-infection within the hygiene management variable. It should be noted that this quadrant contained factors specific to female students, such as a regular diet, for the eating management variable.

In the fourth quadrant (concentrate here), where items show low importance and high performance and receive excessive effort for performance despite their low importance, the following factors are displayed regardless of sex: the meaning of life within the mental health variable, vaccination within the disease management variable, and appropriate diet amount within the eating management variable. It should be noted that this quadrant contained factors specific to male students, such as a regular diet, for the eating management variable.

## 4. Discussion

This study conducted IPA on the health management awareness of 867 middle school students living in the capital area of South Korea during the endemic phase of COVID-19. Two IPAs were completed; one considered sex and the other did not. The analysis revealed the following findings.

The results of IPA on the perception of Korean middle school students regarding the importance of health management awareness and their practice of such behaviors during the current COVID-19 endemic phase compared with the analysis of these behaviors during the COVID-19 pandemic phase are as follows.

First, Korean middle school students placed high importance on the following factors and performed them well during the COVID-19 endemic phase: happiness, interest in life, belongingness, the formation of a reliable relationship, disease prevention, prescription management, and vaccination; regular sports activity, vigorous movement activity, stable sleep environment, habitual hygienic activity, and mandatory hygienic practice. These factors are displayed in the first quadrant (keep up good work) of the IPA matrix, indicating that the performance related to them has been satisfactory and that the maintenance of their current conditions will result in consistently positive results [19,37]. In the early phase of the COVID-19 pandemic, Korean middle school students placed high importance on the variables of disease management, hygiene management, and mental health and exhibited high performance related to these variables [11,38]. The three variables were located in the first quadrant of the IPA matrix, even during the prolonged COVID-19 pandemic phase [25], and this trend was maintained in the COVID-19 endemic phase. Interestingly, the first quadrant of the IPA matrix, focusing on the COVID-19 endemic phase, contains the factors of regular sports activity and vigorous movement activity, unlike the early and prolonged COVID-19 pandemic phases. According to this study’s findings, this difference resulted from the full-scale resumption of non-face-to-face classes in schools and the effects of policies promoting physical education in schools conducted by the Ministry of Education, Offices of Education, and schools to encourage adolescents’ health recovery.

Second, Korean middle school students placed high importance on daily gymnastic activity, regular sleep activity, and appropriate sleep quality but exhibited low performance related to these factors during the COVID-19 endemic phase. These factors are located in the second quadrant of the IPA matrix, where the included items show high importance and low performance, and require immediate performance improvement [20,37]. In contrast, these factors did not appear in the second quadrant of the IPA matrix, which focused on the early and prolonged COVID-19 pandemic phases. Nevertheless, improvements in adolescent performance related to sleep management and physical activity were required during the COVID-19 pandemic period [7,16]. Some studies have shown that the excessive exposure of adolescents to digital media may be the main cause of their failure in sleep management [39,40]. In other words, adolescents recognized the importance of sleep management but exhibited low performance during the COVID-19 endemic phase because of the persistent influence of excessive exposure to digital media and the development of undesirable sleep patterns during the COVID-19 pandemic period.

Third, Korean middle school students placed low importance on the following factors and exhibited low performance during the COVID-19 endemic phase: regular physical activity, appropriate sleep amount, regular diet, ideal eating habits, wearing a facial mask in multi-use facilities, restricted access to multi-use facilities, the practice of distancing, the use of a hand sanitizer, and activity to prevent self-infection. These factors are located in the third quadrant of the IPA matrix, where the included items show low importance and performance and require long-term effort for improvement [31,37]. Other studies showed that, during the early and prolonged COVID-19 pandemic phases, Korean middle school students placed low importance on sleep management, eating management, and physical activity and exhibited poor performance [11,41]. This trend was partially maintained during the COVID-19 endemic phase. However, it is intriguing to note that the third quadrant of the IPA matrix contains the social distancing variable during the COVID-19 endemic phase, and not during the early and prolonged COVID-19 pandemic phases. This difference resulted from a natural shift in the perception and behavioral patterns of Korean middle school students who witnessed the end of social distancing and became aware of the characteristics of weak Omicron variants.

Fourth, Korean middle school students placed low importance on the following factors but exhibited high performance during the COVID-19 endemic phase: the meaning of life, appropriate diet amount, and preventing the infection of others. These factors are located in the fourth quadrant of the IPA matrix, where the included items show low importance and high performance, and receive excessive effort for performance despite their low importance [21,35]. By contrast, these factors were not found in the fourth quadrant of the IPA matrix during the early and prolonged COVID-19 pandemic phases [11,25]. These results imply that Korean middle school students who continuously practiced health management awareness related to the meaning of life were affected by a sense of freedom during the COVID-19 pandemic period, appropriate diet amounts encouraged by the resumed provision of school meals, and infection prevention for others’ benefit that was learned during the COVID-19 pandemic period.

Another study found that several health management awareness factors for Korean middle school students exhibited low performance during the COVID-19 endemic phase: daily gymnastic activity, regular physical activity, regular diet, ideal eating habits, regular sleep activity, appropriate sleep quality, and appropriate sleep amount [42]. Thus, efforts should be made to improve performance related to these factors. In the COVID-19 endemic phase, schools actively encouraged physical activity using a policy-based approach and resumed a school lunch system. This demonstrates a need to provide students with proactive guidance on daily, regular, and ideal health-management practices related to physical activity, eating, and sleep at the family level. Furthermore, social distancing was observed in the third quadrant, showing low importance and performance during the COVID-19 endemic phase. This suggests that Korean middle school students were accurately aware of the current COVID-19 endemic phase and that they carefully reflected on the present conditions in evaluating the importance of social distancing and managing performance related to this variable.

Moreover, the comparison of Korean middle school students’ perception regarding health management awareness during the COVID-19 endemic by sex indicated differences in regular sports and vigorous movement activities within the physical activity variable. Additionally, there was a difference in the regular diet for eating management variables between male and female students. Some studies showed that, during the early and prolonged COVID-19 pandemic phases, both male and female students exhibited low performance in terms of physical activity and eating habits [5,42,43]. However, male students actively practiced regular sports activities, vigorous movement activities, and regular diet during the COVID-19 endemic phase. In contrast, female students maintained a low performance related to these factors during this period. It is difficult to say that this phenomenon was caused by learning effects generated during the COVID-19 pandemic, given the absence of reports on educational inequality between boys and girls in online classes and mixed classes (online and offline learning) during the COVID-19 pandemic period. Rather, this phenomenon may be attributed to a fundamental difference in resilience related to physical activity and eating habits between male and female students, or to the active receptivity of male students to school-based health recovery programs compared with female students during the COVID-19 endemic phase. Based on these results, a more active physical activity recovery and activation program must be designed and applied for female students at school. Careful observations should also determine whether sex inequality has an effect on childcare in the home.

Because this study sampled some middle schools located in a metropolitan area of Korea, the study results are limited and may not represent all middle school students. Follow-up research that can expand the scope of sampling by region and conduct a comparative analysis with the results of this study is necessary.

## 5. Conclusions

This study conducted IPAs based on Korean middle school students during the COVID-19 endemic phase to understand the perception of Korean middle school students toward health management awareness by sex. To this end, Korean middle school students participated in online and offline surveys between May and July 2023. The analysis presents several findings, as follows.

The analysis of the IPA matrix based on Korean middle school students, regardless of sex, indicates that performing regular sports activity and vigorous movement activity increased after the COVID-19 pandemic period ended. However, factors related to social distancing exhibited lower importance and performance during that period. The analysis of the IPA matrix by sex revealed that male students exhibited improved performance in regular sports activity, vigorous movement activity, and regular diet compared to female students during the COVID-19 endemic phase; however, during the COVID-19 pandemic period, both male and female students exhibited low performance for the same factors.

Based on the analysis, this study presents the following conclusions.

First, health recovery programs in schools during the COVID-19 endemic phase had positive effects on middle school students in terms of physical activity, eating habits, and mental management. However, as they showed low performance in daily and regular physical activity, sleep activity, and eating management at home, measures for solving this issue should be developed. Furthermore, differences in physical activity and eating habits were observed between male and female students during the COVID-19 pandemic. These results suggest the need to constantly promote physical activity in female students and provide them with education on eating habits. Additionally, it is necessary to monitor the sex inequality in these factors between male and female students.

In future research, various health management analyses based on a wider range of participants during the COVID-19 endemic phase should be conducted. Moreover, the findings of this study can be a comparison point for future studies from the perspectives of country, region, race, and sex equality. Qualitative research using in-depth interviews or participant observations can be implemented to examine those currently in the recovery stage.

## Figures and Tables

**Figure 1 healthcare-12-00763-f001:**
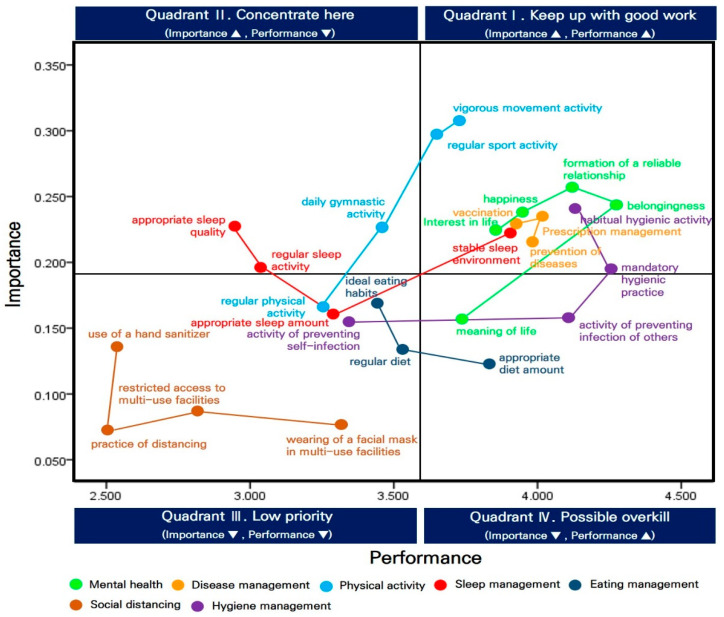
The IPA matrix based on the perception of Korean middle school students toward health management awareness regardless of sex.

**Figure 2 healthcare-12-00763-f002:**
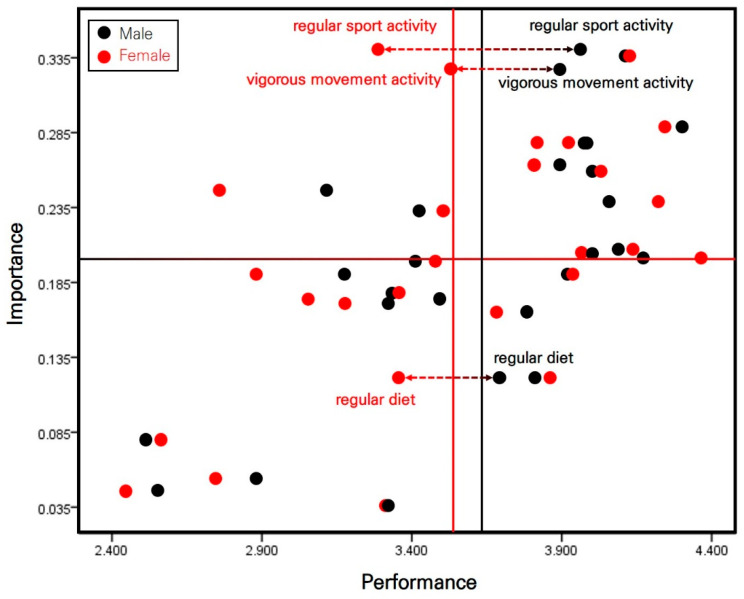
The IPA matrix based on the perception of Korean middle school students toward health management awareness by sex.

**Table 1 healthcare-12-00763-t001:** The demographic characteristics of the research participants.

**Sex**	**Classification**	**Number**	**Percentage (%)**
	Male	464	53.5
	Female	403	46.5
Total		867	100

**Table 2 healthcare-12-00763-t002:** Exploratory factor analysis and reliability analysis.

Variable	Mental Health	Sleep Management	Hygiene Management	Physical Activity	Social Distancing	Disease Management	Eating Management
interest in life	0.812	0.170	0.134	0.088	0.079	0.108	0.106
happiness	0.797	0.174	0.141	0.063	0.054	0.071	0.085
formation of a reliable relationship	0.724	0.128	0.117	0.151	−0.065	0.222	0.041
belongingness	0.722	0.118	0.089	0.098	−0.105	0.147	0.043
meaning of life	0.675	0.082	0.061	0.193	0.103	0.154	0.124
appropriate sleep amount	0.140	0.815	0.036	0.065	0.093	0.060	0.144
regular sleep activity	0.116	0.725	0.068	0.155	0.133	0.095	0.149
appropriate sleep quality	0.196	0.706	0.158	0.178	0.140	0.089	0.130
stable sleep environment	0.151	0.645	0.021	0.021	−0.002	0.100	0.120
habitual hygienic activity	0.123	0.002	0.832	0.085	0.072	0.050	0.145
mandatory hygienic practice	0.146	0.028	0.793	0.039	0.026	0.024	0.129
activity of preventing infection of others	0.119	0.038	0.651	0.096	0.072	0.172	−0.004
activity of preventing self-infection	0.072	0.190	0.632	0.122	0.119	0.137	0.021
vigorous movement activity	0.192	0.099	0.067	0.823	−0.061	0.057	0.064
regular sports activity	0.172	0.113	−0.007	0.795	−0.091	0.052	0.027
daily gymnastic activity	0.059	−0.041	0.185	0.670	0.081	0.075	0.178
regular physical activity	0.082	0.189	0.111	0.637	0.046	0.021	0.039
practice of distancing	0.053	0.131	0.121	0.050	0.828	0.084	−0.001
restricted access to multi-use facilities	0.039	0.086	−0.002	−0.050	0.751	0.025	0.100
wearing a facial mask in multi-use facilities	−0.019	0.006	−0.012	−0.123	0.724	0.017	0.100
use of a hand sanitizer	−0.038	0.112	0.292	0.186	0.626	0.117	0.009
prescription management	0.233	0.085	0.134	0.082	0.087	0.812	0.132
vaccination	0.161	0.108	0.148	0.046	0.087	0.785	0.176
prevention of diseases	0.258	0.170	0.128	0.078	0.062	0.752	0.098
appropriate diet amount	0.161	0.157	0.128	0.143	0.106	0.100	0.773
ideal eating habits	0.095	0.148	0.126	0.154	0.073	0.190	0.766
regular diet	0.097	0.374	0.042	0.014	0.072	0.126	0.642
Eigenvalue	3.219	2.570	2.479	2.452	2.334	2.129	1.862
Variance (%)	11.923	9.517	9.183	9.080	8.643	7.887	6.897
Cumulative (%)	11.923	21.440	30.623	39.703	48.347	56.233	63.130
Cronbach’s α	0.852	0.787	0.752	0.761	0.739	0.814	0.728

**Table 3 healthcare-12-00763-t003:** The values of relative importance and performance of health management awareness of Korean middle school students regardless of sex.

Variables		Importance	Performance
Mental health	Happiness	0.238	3.950
	interest in life	0.225	3.854
	Belongingness	0.244	4.273
	formation of a reliable relationship	0.257	4.119
	meaning of life	0.157	3.734
Disease management	prevention of diseases	0.216	3.980
	prescription management	0.235	4.016
	vaccination	0.229	3.925
Physical activity	regular sport activity	0.297	3.648
	daily gymnastic activity	0.227	3.459
	vigorous movement activity	0.308	3.727
	regular physical activity	0.166	3.253
Sleep management	regular sleep activity	0.196	3.036
	appropriate sleep quality	0.228	2.947
	appropriate sleep amount	0.161	3.287
	stable sleep environment	0.223	3.905
Eating management	regular diet	0.134	3.532
	ideal eating habits	0.169	3.442
	appropriate diet amount	0.123	3.833
Social distancing	wearing a facial mask in multi-use facilities	0.077	3.317
	restricted access to multi-use facilities	0.087	2.817
	practice of distancing	0.073	2.504
	use of a hand sanitizer	0.136	2.535
Hygiene management	activity of preventing infection of others	0.158	4.107
	activity of preventing self-infection	0.155	3.346
	habitual hygienic activity	0.241	4.130
	mandatory hygienic practice	0.195	4.255
Average		0.191	3.590

**Table 4 healthcare-12-00763-t004:** The quadrants of the IPA matrix based on the perception of Korean middle school students toward health management awareness regardless of sex.

Quadrant	Criteria	Variables Distribution
Quadrant I	Importance ↑ Performance ↑	happiness, interest in life, belongingness, formation of a reliable relationship, prevention of diseases, prescription management, vaccination, regular sports activity, vigorous movement activity, stable sleep environment, habitual hygienic activity, mandatory hygienic practice
Quadrant II	Importance ↑ Performance ↓	daily gymnastic activity, regular sleep activity, appropriate sleep quality
Quadrant III	Importance ↓ Performance ↓	regular physical activity, appropriate sleep amount, regular diet, ideal eating habits, wearing of a facial mask in multi-use facilities, restricted access to multi-use facilities, practice of distancing, use of a hand sanitizer, activity of preventing self-infection
Quadrant IV	Importance ↓ Performance ↑	meaning of life, appropriate diet amount, activity of preventing infection of others

above average: ↑, below average: ↓.

**Table 5 healthcare-12-00763-t005:** Difference in the sex of Korean middle school students for health management awareness by sex.

Categories	Sex	N	M	SD	SE	*t*	*p*
Mental health	Male	464	4.012	0.750	0.035	1.103	0.270
	Female	403	3.956	0.746	0.037		
Disease management	Male	464	3.971	0.834	0.039	−0.126	0.900
	Female	403	3.978	0.830	0.041		
Physical activity	Male	464	3.649	0.959	0.045	4.331	0.000 ***
	Female	403	3.375	0.898	0.045		
Sleep management	Male	464	3.439	1.000	0.046	4.669	0.000 ***
	Female	403	3.127	0.964	0.048		
Eating management	Male	464	3.636	0.965	0.045	1.147	0.252
	Female	403	3.562	0.928	0.046		
Social distancing	Male	464	2.817	0.983	0.046	0.802	0.423
	Female	403	2.766	0.919	0.046		
Hygiene management	Male	464	3.910	0.803	0.037	−2.058	0.040 *
	Female	403	4.017	0.723	0.036		

*** *p* > 0.001, * *p* > 0.05.

**Table 6 healthcare-12-00763-t006:** The values of relative importance and performance based on the perception of Korean middle school students toward health management awareness by sex.

Variables		Male Students		Female Students	
		Importance	Performance	Importance	Performance
Mental health	Happiness	0.278	3.976	0.278	3.921
	interest in life	0.263	3.894	0.263	3.806
	belongingness	0.289	4.300	0.289	4.243
	formation of a reliable relationship	0.336	4.110	0.336	4.129
	meaning of life	0.166	3.780	0.166	3.680
Disease management	prevention of diseases	0.204	3.996	0.204	3.963
	prescription management	0.259	4.002	0.259	4.032
	vaccination	0.191	3.914	0.191	3.938
Physical activity	regular sport activity	0.340	3.961	0.340	3.288
	daily gymnastic activity	0.233	3.420	0.233	3.504
	vigorous movement activity	0.328	3.894	0.328	3.533
	regular physical activity	0.171	3.321	0.171	3.174
Sleep management	regular sleep activity	0.191	3.175	0.191	2.876
	appropriate sleep quality	0.247	3.112	0.247	2.757
	appropriate sleep amount	0.174	3.491	0.174	3.052
	stable sleep environment	0.278	3.978	0.278	3.821
Eating management	regular diet	0.122	3.688	0.122	3.352
	ideal eating habits	0.199	3.412	0.199	3.476
	appropriate diet amount	0.121	3.810	0.121	3.859
Social distancing	wearing of a facial mask in multi-use facilities	0.037	3.321	0.037	3.313
	restricted access to multi-use facilities	0.055	2.879	0.055	2.744
	practice of distancing	0.046	2.556	0.046	2.444
	use of a hand sanitizer	0.080	2.513	0.080	2.561
Hygiene management	activity of preventing infection of others	0.207	4.086	0.207	4.132
	activity of preventing self-infection	0.178	3.336	0.178	3.357
	habitual hygienic activity	0.239	4.056	0.239	4.216
	mandatory hygienic practice	0.202	4.162	0.202	4.362
Average		0.201	3.635	0.201	3.538

**Table 7 healthcare-12-00763-t007:** The four quadrants of the IPA matrix based on the perception of Korean middle school students toward health management awareness by sex.

Quadrant	Criteria	Sex	Variables Distribution
Quadrant I	Importance ↑ Performance ↑	Male	happiness, interest in life, belongingness, formation of a reliable relationship, prevention of diseases, prescription management, regular sports activity, vigorous movement activity, stable sleep environment, activity of preventing infection of others, habitual hygienic activity, mandatory hygienic practice
		Female	happiness, interest in life, belongingness, formation of a reliable relationship, prevention of diseases, prescription management, stable sleep environment, activity of preventing infection of others, habitual hygienic activity, mandatory hygienic practice
Quadrant II	Importance ↑ Performance ↓	Male	daily gymnastic activity, appropriate sleep quality
		Female	regular sport activity, daily gymnastic activity, vigorous movement activity, appropriate sleep quality
Quadrant III	Importance ↓ Performance ↓	Male	regular physical activity, regular sleep activity, appropriate sleep amount, ideal eating habits, wearing of a facial mask in multi-use facilities, restricted access to multi-use facilities, practice of distancing, use of a hand sanitizer, activity of preventing self-infection
		Female	regular physical activity, regular sleep activity, appropriate sleep amount, regular diet, ideal eating habits, wearing of a facial mask in multi-use facilities, restricted access to multi-use facilities, practice of distancing, use of a hand sanitizer, activity of preventing self-infection
Quadrant IV	Importance ↓ Performance ↑	Male	meaning of life, vaccination, regular diet, appropriate diet amount
		Female	meaning of life, vaccination, appropriate diet amount

above average: ↑, below average: ↓.

## Data Availability

The data presented in this study are available upon request from the corresponding author. The data were not publicly available to ensure protection of personal information.

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
