# Peer review of "An Importance-Performance Analysis of Korean Middle School Students’ Health Management Awareness during the Post-COVID-19 Era Depending on Sex"

_healthcare, 2024, doi:10.3390/healthcare12070763_

Round 1

Reviewer 1 Report

Comments and Suggestions for Authors

Major comments:

The IPA method is rarely presented in public health journals. I think that few readers are familiar with it, which makes it difficult to perceive the text. Please describe in more detail in the introduction its principles and benefits in the context of your topic. Also in the methods section, it would be useful to describe in more detail what the two IPA indicators are and what the graphical presentation of the results is.  Figure 1 could be labelled with standard quadrant names (such as possible overkill) to make it easier to associate with the description in the text.

The introduction should relate to all subject areas and is dominated by the theme of physical activity. The concept of health management has not been defined.

The description of the instrument and its source is very vague. The original source reference [25] contains a 32-question questionnaire with other dimensions. It is difficult to guess whether this is a well-known tool or built ad hoc from a generic health management model. I recommend including the actual questionnaire in the electronic appendix. The tables show that there are 28 variables indexed in 7 dimensions. I suggest standardising the terminology, avoiding the word variable, and calling the questions items and the dimensions factors/dimensions. It should be clear from the description of the survey instrument and the annex how performance and importance are measured (what ranges they have) and what the numbers mean.

The abstract is incomprehensible to a reader unfamiliar with the IPA. It is better not to use the quadrant numbers themselves, but their usual names. In the discussion, too, the numbers of the quadrants were better supplemented by their names.

The discussion and conclusions contain too many repetitions of results. Only three new papers are cited (31-33), including two Korean.  There are many studies on how students functioning during a pandemic.

It is very unclear to refer to previous studies and to treat this study as a follow up, as previous results are not discussed. Equally unclear is the discussion of the results in the context of ongoing programmes.

Minor comments.

The title of Table 2 is too general. Are there factor loadings from EFA? It would be useful to bold the items corresponding to each factor.

By the sentence starting at line 88, there is no reference to literature. The same applies to the sentence starting on line 96.

There is a bookmark error in the name of Figure 1.

Figure 1 should have an explanation of colours.

Item 25 is cited after 26, 27 and 28.

The literature is double numbered.

Author Response

Thank you very much for your kind suggestion and recommendation.

Reviewer 2 Report

Comments and Suggestions for Authors

General Comment

Overall, this is a well-conducted study analyzing an important research question on the health perceptions and behaviors of adolescents during the post-COVID transition period. The results highlighting differences compared to the pandemic period as well as disparities between male and female students are interesting. The study makes a useful contribution to the literature.

Specific Comments

Abstract

  • The abstract provides a nice concise summary of the study background, methods, results, and conclusions.
  • The background establishes the rationale and objectives clearly.
  • The methods overview is good, but consider adding 1-2 more details like the date range of data collection.
  • The results are summarized well with key findings highlighted.
  • The conclusions effectively summarize the main implications.

Introduction

  • The introduction provides good background and rationale for the study aim.
  • The flow and organization of information is logical.
  • References are appropriately cited to support statements.
  • Consider expanding a bit more on the knowledge gaps this study aims to address. What key questions remain unanswered that this study will provide insights on?
  • When reviewing previous literature, highlight what is known versus unknown to build the case for this study.

Materials and Methods

  • The overall description of the study design, participants, data collection, and analysis is clear.
  • Consider providing more details for some aspects to aid reproducibility:
    • Expand on the survey development process - was it adopted from previous surveys? How were specific questions and scales determined?
    • Provide more information about the statistical analysis methods - statistical tests used, any correction for multiple testing, software and tools used.
  • For participants:
    • Describe how the sample size was determined. Was power analysis performed?
    • Provide info about inclusion/exclusion criteria if any.
  • For data collection:
    • Describe the survey administration in more detail - online vs paper, settings, time taken, etc.
  • For data analysis:
    • Describe any missing data handling or quality control measures.

Discussion

  • The discussion of results in the context of previous literature is good.
  • Consider expanding more on the implications and applications of your findings. For example:
    • What are the specific strategies that can be implemented based on your results?
    • How can the disparities found between male and female students be addressed?
  • Elaborate on the limitations of the study:
    • Discuss limitations of convenience sampling and self-reported data.
    • How might biases have been introduced and how can they be minimized in future studies?
  • Suggest potential future research directions to build on your work.
  • Make sure the conclusions are supported by the results presented.
  • Avoid overstating the implications of your findings.
Comments on the Quality of English Language
  • Overall, the English language is good and easy to understand. There are only minor issues in some parts.
  • There are a few grammatical errors - misuse of articles, prepositions, plural vs singular, etc. Recommend proofreading thoroughly.
  • Some sentences are lengthy or awkwardly structured. Break these into shorter, clearer sentences.
  • Be consistent in terminology and vocabulary usage, e.g. use either 'adolescents' or 'middle school students' consistently instead of interchangeably.
  • Some parts contain repetitive statements or redundant information. This can be condensed.
  • Check for consistent formatting and style - spacing, fonts, etc.
  • Make sure references are formatted properly and consistently.
  • Watch for typos, spelling errors, or missing words.

Author Response

(The authors gave the same response as above.)
